# Ultra-Compact and Broadband Nano-Integration Optical Phased Array

**DOI:** 10.3390/nano13182516

**Published:** 2023-09-08

**Authors:** Zhicheng Wang, Junbo Feng, Haitang Li, Yuqing Zhang, Yilu Wu, Yuqi Hu, Jiagui Wu, Junbo Yang

**Affiliations:** 1College of Artificial Intelligence, Southwest University, Chongqing 400715, China; zcwang1027@foxmail.com (Z.W.); zhangyuqing594@163.com (Y.Z.); imwuyilu@163.com (Y.W.); hyq895651004@163.com (Y.H.); 2Center of Material Science, National University of Defense Technology, Changsha 410073, China; 3United Microelectronics Center Co., Ltd., Chongqing 401332, China; junbo.feng@cumec.cn; 4School of Physical Science and Technology, Southwest University, Chongqing 400715, China; lht15223540254@163.com

**Keywords:** optical phased array, inverse design, optical power splitter

## Abstract

The on-chip nano-integration of large-scale optical phased arrays (OPAs) is a development trend. However, the current scale of integrated OPAs is not large because of the limitations imposed by the lateral dimensions of beam-splitting structures. Here, we propose an ultra-compact and broadband OPA beam-splitting scheme with a nano-inverse design. We employed a staged design to obtain a T-branch with a wavelength bandwidth of 500 nm (1300–1800 nm) and an insertion loss of −0.2 dB. Owing to the high scalability and width-preserving characteristics, the cascaded T-branch configuration can significantly reduce the lateral dimensions of an OPA, offering a potential solution for the on-chip integration of a large-scale OPA. Based on three-dimensional finite-difference time-domain (3D FDTD) simulations, we demonstrated a 1 × 16 OPA beam-splitter structure composed entirely of inverse-designed elements with a lateral dimension of only 27.3 μm. Additionally, based on the constructed grating couplers, we simulated the range of the diffraction angle θ for the OPA, which varied by 0.6°–41.6° within the wavelength range of 1370–1600 nm.

## 1. Introduction

Compared to traditional mechanical LiDAR modules, optical phased arrays (OPAs) offer advantages such as compact size, fast and stable beam control, low cost, and low power consumption, making them widely applicable in various fields, including LiDAR [1,2,3], image projection [4,5,6], and free-space optical communication [7,8,9]. In recent years, with advancements in semiconductor processes and the development of silicon-on-insulator (SOI) technology, silicon waveguide-based OPAs have made significant progress in various aspects such as different operating wavelength [10,11,12], antenna aperture size [13,14,15], and wide scanning range [16,17,18]. However, reported OPAs are mostly limited to smaller scales. Owing to the complexity of phase modulation and the size issues caused by component cascading, the integration of large-scale OPAs still faces challenges.

Optical power splitters are important components of OPA. The cascaded power splitter, phase shifters, and antenna array occupy the majority of the OPA’s area. Common on-chip splitters include Y-branch splitters and multimode interference couplers (MMI) [19,20,21,22]. However, as the array size increased, the lateral dimensions of the bent waveguides connecting the adjacent Y-branch or MMI components also increased, resulting in a larger lateral dimension for the entire OPA. T-branch power splitters have unique advantages in this respect. Regardless of the array size, the lateral dimensions of the waveguides connecting adjacent T-branches are always fixed. However, current T-branch beam splitters [23,24,25] have a large insertion loss and are difficult to use in the beam splitter structure of OPA. An inverse design starts with optimization objectives and utilizes relevant intelligent algorithms [26,27,28] to optimize multiple design parameters, thereby enabling the determination of the required optical structures. As the inverse design transforms the device design into the optimization of multiple parameters without being restricted to a specific structure, it offers greater design flexibility, often resulting in more compact and higher-performing devices [29,30,31,32,33,34,35]. For example, Huang et al. [29] applied an objective-first method to obtain a dual-channel wavelength demultiplexer with a size of only 2.4 × 10 μm^2^. Tyler et al. [31] designed a compact nonlinear optical switch using an adjoint method. Xu et al. [33] combined topological optimization and a direct binary algorithm to achieve a dual-mode ultra-broadband power splitter. Therefore, the inverse design provides a new solution for obtaining a more compact large-scale OPA.

Shape adjoint optimization [36], as an intelligent optimization algorithm in inverse design, requires only one forward simulation and one adjoint simulation to obtain the corresponding shape gradient information, making its computational cost relatively low. By introducing the gradient descent method, the positions of boundary points are iteratively changed to achieve shape optimization. Currently, the shape-adjoint optimization algorithm has been applied to various optical devices. For instance, Ruan et al. [37] proposed a two-level shape-adjoint method, based on which they obtained a compact polarization beam-splitting rotator with a footprint of only 3 × 17 μm, featuring low crosstalk and low loss. Piggott et al. [38], with the addition of appropriate manufacturing constraints, achieved a low loss, compact 1 × 3 power splitter with a bandwidth of 300 nm.

In this paper, we propose an ultra-compact and broadband OPA design based on an inverse-designed T-branch. The T-branch power splitter is obtained through a staged shape-adjoint optimization design, with a range of variation in insertion loss of −0.2 dB within the wavelength range of 1300–1800 nm. The lateral dimension of the T-branch is 7 μm, and the longitudinal dimension can be adjusted based on the size of the OPA array. We demonstrated a 1 × 16 OPA array composed of inverse-designed T-branches and Y-branches with a spacing of 2 μm between the waveguides. The antenna array employed a width-modulated grating coupler for emission. In the 1370–1600 nm wavelength range, the variation range of the diffraction angle θ is 41°.

## 2. Design and Optimization

Figure 1a depicts our 1 × 16 ultra-compact OPA scheme based on an inverse design. The experimental simulation uses the 3D FDTD method, and the software is Lumerical FDTD 2020 R2. We conducted simulation tests on a 220 nm-thick silicon-on-insulator (SOI) platform. The simulation grid size is set to 30 nm. The incident light source is configured as the fundamental TE mode, with a wavelength range from 1300 nm to 1800 nm. The x-direction simulation boundary is set to perfectly matched layers (PML), the y-direction boundary is anti-symmetric, and the z-direction boundary is symmetric. The simulation time is set to 5000 fs.

For the entire beam-splitting structure, we employed four inverse-designed beam splitters. For the third and fourth layers with a larger vertical spacing, we utilized inverse-designed T-branch waveguides. For the first and second layers with smaller vertical spacings, we employed inverse-designed Y-branch waveguides, as illustrated in Figure 1b. Figure 1c shows a conventional OPA beam-splitting structure composed of cascaded MMIs or Y-branches. The lateral spacing L_n_ between the splitters can be expressed as follows:(1)$$L_{n}=k\times \left[\sum_{i=1}^{n-1}h_{i}+g-w\times \left(n-1\right)\right]+l_{couple}$$

Therefore, the overall lateral size of the OPA after cascading the MMI or Y-branch is
(2)$$sum\left(L_{n}\right)=\sum_{i=1}^{n}L_{i}$$

In Equation (1), k represents the ratio between the curved part of the lateral spacing and the corresponding height. The results in square brackets represent the height of the splitter in each layer. $\sum_{i=1}^{n-1}h_{i}$ represents the sum of the heights of the splitters from the 1st layer to the (n − 1)th layer, *g* represents the spacing between adjacent splitters in the first layer, w represents the width of the waveguide, and $l_{couple}$ represents the coupling length of the splitter. Here, we set k = 1.1, g = 2 μm, w = 0.5 μm, and $l_{couple}$ = 12 μm. Thus, as the array size increases, the lateral spacing *L_n_* between adjacent devices also increases. This resulted in a larger cascaded structure. For the inverse-designed T-branch waveguides herein proposed, the lateral spacing remains constant at 7 μm regardless of the increase in array size. The lateral size of the cascaded OPA is expressed as follows:(3)$$sum\left(L_{n}^{\prime }\right)=7\times n$$

This significantly reduces the lateral size of traditional OPAs, thus facilitating the on-chip integration of large-scale OPAs.

The four optical power splitters shown in Figure 1b were obtained using a staged shape-adjoint optimization design. As a form of the adjoint method in inverse design, shape-adjoint optimization establishes a mapping relationship between the coordinates of the boundary points (*x* and *y*). We use the python API in Lumerical FDTD 2020 (which is described in detail on the official website and is open source) to pass custom shape functions, material properties, optimized geometric ranges and constraints, and then perform iterative optimization according to the algorithm flow diagram shown in Figure 2a.

In order to make the optimized boundary as smooth as possible, we set the relationship between the boundary point coordinates *x* and *y* as cubic interpolation, that is, the multi-piecewise third-order function S(x). Specifically, it can be interpreted as under the condition of given n + 1 boundary points, there are n intervals. For each segment interval [$x_{i}$, $x_{i+1}$] (i = 0, 1, …, n − 1), $S_{i}\left(x\right)$ are both cubic polynomials and satisfy the conditions of Equation (4).
(4)$$S\left(x_{i}\right)=y_{i}$$

Furthermore, for *S*(*x*), its first derivative *S’*(*x*), and second derivative *S”*(*x*), they are all continuous within the given range of boundary points. Thus, n cubic polynomials can be represented piecewise as follows:(5)$$S_{i}\left(x\right)=a_{i}+b_{i}\left(x-x_{i}\right)+c_{i}{\left(x-x_{i}\right)}^{2}+d_{i}{\left(x-x_{i}\right)}^{3}$$

Here, $a_{i}$, $b_{i}$, $c_{i}$, and $d_{i}$ represent the 4n unknowns to be solved. For a detailed derivation, please refer to reference [39]. By continuously modifying the *y*-coordinate values of the boundary points based on the gradient information obtained from forward and adjoint simulations and through iterative optimization, the shape is obtained with convergence of the figure of merit (FOM). The FOM is defined as Equation (6):(6)$$FOM=\frac{\sum_{\lambda_{a}}^{\lambda_{b}}T_{up}\left(\lambda_{i}\right)+\sum_{\lambda_{a}}^{\lambda_{b}}T_{down}\left(\lambda_{i}\right)}{N}$$

$T_{up}$ represents the transmittance value of the upper channel on the output waveguide, while $T_{down}$ represents the transmittance value of the lower channel on the output waveguide. $\lambda_{i}$ represents the wavelength within the [$\lambda_{a}$, $\lambda_{b}$] interval. *N* represents the number of wavelengths $\lambda_{i}$. A larger *N* allows for a finer division of $\lambda_{i}$, resulting in a more accurate FOM value but requiring longer simulation times. Here, we set $\lambda_{a}$ = 1300 nm, $\lambda_{b}$ = 1800 nm, and N = 50.

Figure 2b–d illustrates the optimization process of the first stage. Initially, a Euler waveguide [40,41,42] was used as the output waveguide for the T-branch, and through shape-adjoint optimization, the optimized structure of the T-branch coupling was obtained, as shown in Figure 2d. The second stage involves using the optimized coupling structure obtained in the first stage as the initial structure and further applying an inverse design to achieve a more compact and lower insertion loss output waveguide for the T-branch. The optimization process for the output waveguide is illustrated in Figure 2f–h. The two inverse-designed Y-branch structures had similar optimization processes. The phased optimization of the coupling region and the curved waveguide region takes an average of 10 min per iteration on a server equipped with 28 core computing units and 128 GB of memory (equipped with Intel(R) Xeon(R) CPU E5-2680 v4 of 2.4 GHz). The number of iterations required was 30 and 46, respectively. The total optimization time does not exceed 13 h.

## 3. Simulation and Results

Figure 3a,d illustrate the structures of the two types of T-branch waveguides obtained through staged-shape adjoint optimization. The widths of the input and output waveguides for both T-branch designs were set to 500 nm. L1 and L2 are the lengths of the straight waveguides in the T-branch structure. As the array size increases, the T-branch requires only adjusting the length of the straight waveguide to stretch its longitudinal length. The width of the straight waveguide was set to 800 nm to provide a wider turning range for the light path. The lower bent waveguide connected to the straight waveguide was designed using an inverse design approach, whereas the upper half of the output waveguide was obtained by rotating the lower half. For the T-branch in the fourth layer of the OPA beam splitter structure, H1 = 20.5 μm and W1 = 7 μm. For the T-branch in the third layer, H2 = 10.5 μm and W2 = 7 μm. Figure 3b,e show the optical-field distributions of the two T-branches at 1550 nm. Figure 3c,f show the insertion loss curves, where the red curve represents the overall insertion loss of the T-branches. The variation range of the insertion loss for both T-branches is −0.2 dB within the wavelength range of 1300–1800 nm. Therefore, using a staged inverse design, we achieved a highly scalable ultra-broadband and compact T-branch power splitter.

To validate the scalability and suitability of our proposed T-branch for large-scale OPA, we simulated five different heights of T-branches, representing the outermost T-branches in the 1 × 32, 1 × 64, 1 × 128, 1 × 256, and 1 × 1024 OPAs. Their heights H are 40.5 μm, 80.5 μm, 160.5 μm, 320.5 μm, and 1280.5 μm, respectively. Figure 4a–c depict the optical field distributions of the outermost T-branches of the OPA at 1550 nm on a certain scale. Additionally, we compared the insertion loss of the five different-height T-branches with the outermost T-branch of the 1 × 16 OPA, as shown in Figure 4d. For the proposed two-stage inverse-designed T-branch, when the OPA scale changes, only the length of the straight waveguide in the T-branch must be adjusted. Therefore, when the OPA scale does not exceed 1 × 256, the range of variation in the insertion loss in the T-branch within the wavelength range of 1300–1800 nm is within −0.2 dB. Even when the OPA scale reaches 1 × 1024, the insertion loss in the T-branch still remains within a variation range of −0.2 dB.

In addition, we compared the T-branch of the one-stage inverse design with that of the two-stage inverse design, as shown in Figure 5. The structures within the yellow dashed boxes were all obtained through inverse design. It can be observed that the boundaries of these optimized structures vary, which is a characteristic of shape-adjoint optimization in inverse design. Figure 5a illustrates the T-branch subjected to a one-stage inverse design, where we employed low-loss Euler waveguides as the bent waveguides of the T-branch and optimized the shape of its coupling region through an inverse design. However, because of the fixed bending radius of Euler waveguides at a certain curvature radius, they may not be suitable for T-branches with arbitrary stretching lengths, which can lead to light leakage, as shown in Figure 5b. Figure 5c shows the T-branch obtained using the two-stage inverse design, and Figure 5d shows the optical field distribution at a wavelength of 1550 nm. The T-branch obtained through the two-stage inverse design not only has a lower insertion loss at the bent waveguide but also reduces the lateral dimensions by 2.82 μm. Figure 5e compares the variation range of the insertion loss for the two types of T-branch structures in the wavelength range of 1300–1800 nm. The T-branch obtained through the two-stage inverse design exhibited significantly better performance than the T-branch obtained through the one-stage inverse design, particularly in the wavelength range of 1500–1800 nm.

Both traditional and reverse-designed T-branch devices have been reported to exhibit excellent performance. Here, we have compared their various performance metrics, as shown in Table 1. In addition, the OPA splitter structure includes a large number of curved waveguides, so it is valuable to include vertically height-variable curved waveguides in the comparison.

As the beam-splitter structure approaches the transmission antenna of the OPA, the longitudinal length of the splitter decreases. Therefore, in the first and second layers of the OPA, more compact Y-branch power splitters were used. Figure 6a,d illustrate the Y-branch waveguides obtained through a staged reverse design. The inverse design region is divided into two parts: the coupling region and the curved waveguide. The coupling regions of the two Y-branch waveguides have widths of 2 μm. The height H3 of the Y-branch in the second layer of the OPA beam-splitter structure is 5.5 μm, and the width W3 of the curved waveguide is 6 μm. For the Y-branch in the first layer, its height H4 is 3 μm, and the width W4 of the curved waveguide is 3.3 μm. Figure 6b,e show the optical field distributions of the two Y-branches at a wavelength of 1550 nm. Figure 6c,f, respectively, show the insertion loss curves for the two Y-branches over a wavelength range of 1200–1800 nm, with a variation range of −0.4 dB.

In addition, we also discussed the fabrication tolerance of the splitters [29,46]. Taking the first-layer T-branch and the fourth-layer Y-branch shown in Figure 1a as an example, we simulated the cases of boundary point expansion (+) and contraction (−), and the transmission losses are shown in Figure 7. Our proposed T-branch and Y-branch maintain their original performance even under a fabrication tolerance of “±10 nm”. This reflects the robustness of the devices we proposed. When the fabrication error reaches 20 nm, it has some impact on certain wavelength bands of the T-branch. Interestingly, the performance of the Y-branch is significantly better when the boundary point is shifted by “+20 nm” compared to “−20 nm”. One reasonable explanation is that the “+20 nm” fabrication error, relative to the “−20 nm” error, better preserves the integrity of the original structure, which is beneficial for maintaining the integrity of electromagnetic field propagation.

Based on the ultra-compact OPA power-splitter structure mentioned above, we constructed an optical grating coupler based on the width perturbations, as shown in Figure 8a. The coupling length L = 20 μm, the waveguide spacing d = 1.5 μm, the waveguide width w_1_ = 0.5 μm, the disturbance width w_2_ = 0.8 μm, the grating period Λ = 0.66 μm, and the duty cycle is 50%. When the input wavelength λ changes, the diffraction angle θ also varies. The T-branch designed using a two-stage inverse design process has a bandwidth of 500 nm. Therefore, theoretically, the diffraction angle θ of the proposed OPA has a wide scanning range. Figure 8b depicts the variation in the diffraction angle θ within the wavelength range of 1370–1600 nm, with the range of 0.6°–41.6°. The tuning efficiency was calculated as 0.178°/nm. Figure 8c–h illustrate the far-field distribution at wavelengths λ = 1370 nm, 1400 nm, 1450 nm, 1500 nm, 1550 nm, and 1600 nm, respectively.

To highlight the advantages of our ultra-compact OPA design, we compared it with a traditional cascaded structure using MMI or Y-branch components, as shown in Figure 9. According to Equation (2), as the scale of the OPA array increases, the lateral size based on the MMI or Y-branch scheme exhibits an accumulative trend. When L_n_ reaches 12 (corresponding OPA scale is 1 × 4096), the length of L_12_ alone exceeds 5.5 mm. In contrast, when using our inverse-designed T-branch scheme, regardless of the array size, the lateral spacing L_n_ between adjacent splitters remains 7 μm, as shown in Figure 9a. The inset shows the result of using a logarithmic scale for the *y*-axis. Under the same OPA height conditions, we also compared the areas obtained using different splitting schemes, as shown in Figure 9b. When the OPA reaches a scale of 1 × 4096, the area of the traditional OPA approaches 120 mm^2^, making it difficult to integrate it onto a chip. However, the OPA based on our scheme has an area of less than 1 mm^2^ on the same scale. Compared with the traditional solution, we reduced the area by a factor of 120.

## 4. Conclusions

In conclusion, we applied a shape-adjoint optimization algorithm to a nano-inverse design to obtain an ultra-broadband, low insertion loss, and compact T-branch, based on which we propose an ultra-compact and broadband OPA scheme. In the simulated 1 × 16 OPA beam-splitter structure, the beam splitter of each layer was obtained. A comparison reveals that the T-branches obtained using the two-stage inverse design are more compact and exhibit lower insertion losses than those based on the Euler waveguide. In addition, we also constructed a grating-coupling structure within the wavelength range of 1370–1600 nm. The deflection angle range of θ is between 0.6° and 41.6°. The lateral distance between adjacent T-branches remains constant regardless of the array size of OPA. Therefore, compared with traditional OPA schemes, the proposed nano-T-branch cascaded splitting approach can significantly compress the lateral dimensions of the OPA. This is of great significance for the on-chip integration of large-scale OPAs.

## Figures and Tables

**Figure 1 nanomaterials-13-02516-f001:**
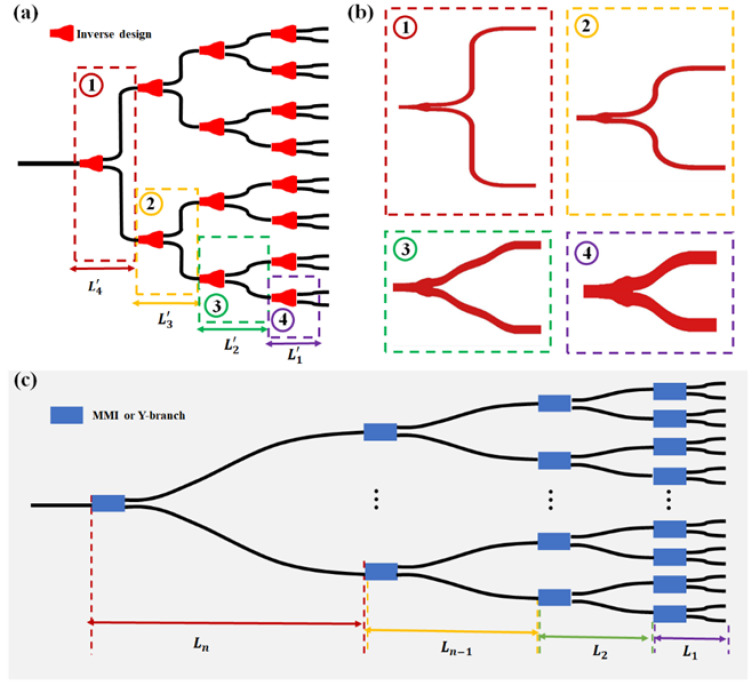
(**a**) Inverse-designed ultra-compact OPA scheme. (**b**) Four inverse-designed beam splitter structures. (**c**) Traditional OPA beam-splitter structure.

**Figure 2 nanomaterials-13-02516-f002:**
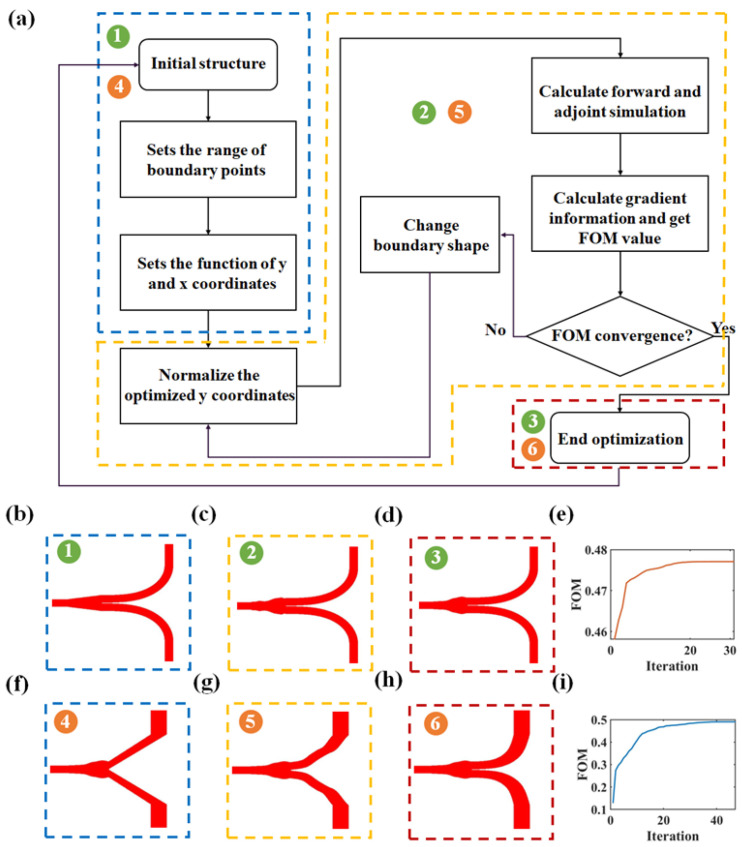
(**a**) Inverse design of the T-branch algorithm flow chart. (**b**–**d**) Optimization process change diagram of T-branch coupling region, and (**e**) iterative change curve of FOM values. (**f**–**h**) Optimization process change diagram of the T-branch output waveguide and (**i**) iterative change curve of FOM value.

**Figure 3 nanomaterials-13-02516-f003:**
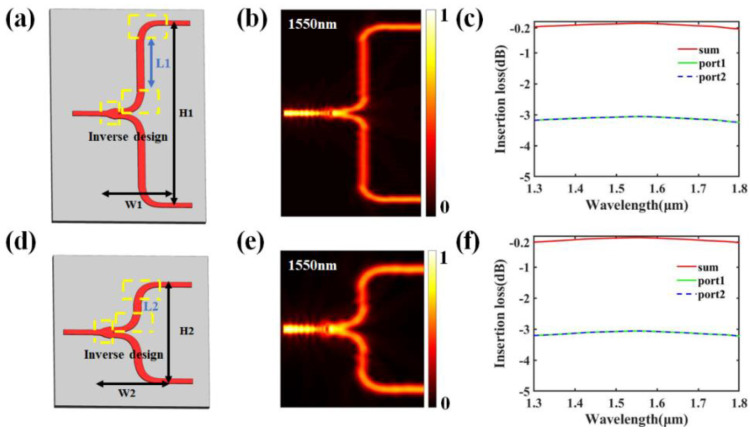
(**a**) T-branch structure diagram of layer four in 1 × 16 OPA. (**b**) Optical field distribution diagram at 1550 nm. (**c**) Simulated insertion loss curve. (**d**) T-branch structure diagram of layer three in 1 × 16 OPA. (**e**) Optical field distribution diagram at 1550 nm. (**f**) Simulated insertion loss curve.

**Figure 4 nanomaterials-13-02516-f004:**
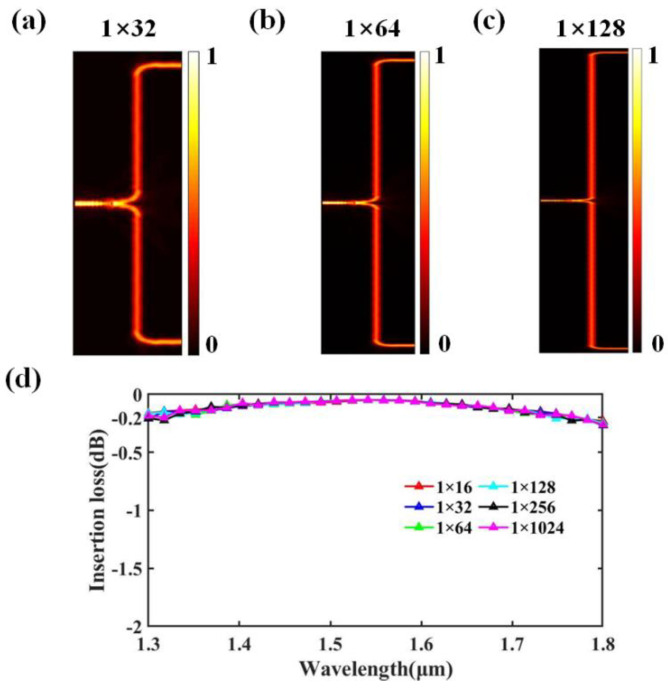
Optical field distribution of the outermost T-branch in (**a**) 1 × 32 OPA, (**b**) 1 × 64 OPA, and (**c**) 1 × 128 OPA. (**d**) Comparison curves of insertion loss in T-branches with different heights.

**Figure 5 nanomaterials-13-02516-f005:**
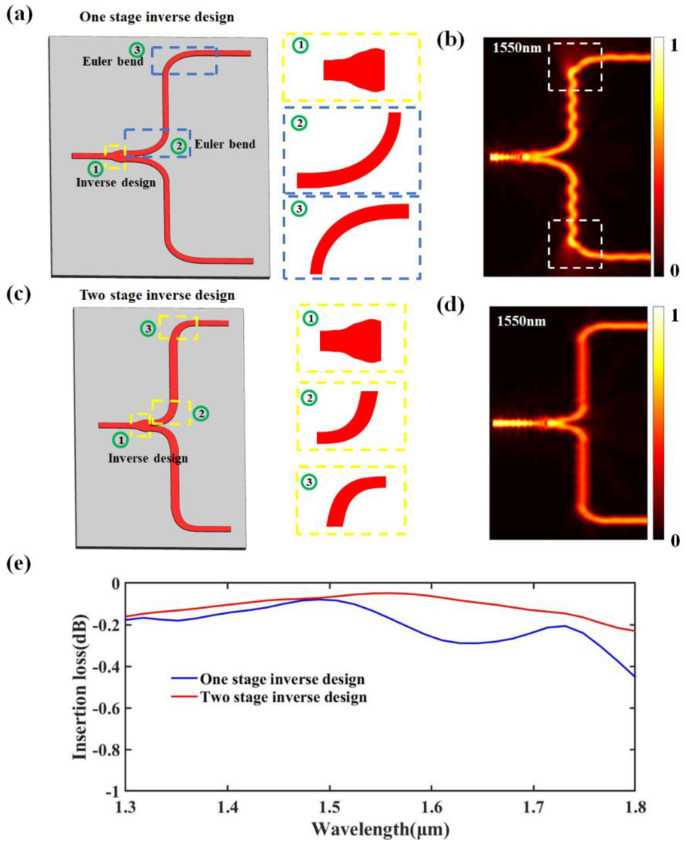
Schematic diagram of the T-branches obtained through (**a**) one-stage inverse design and (**c**) two-stage inverse design. (**b**,**d**) Optical field distribution at 1550 nm. (**e**) Comparison of insertion loss curves.

**Figure 6 nanomaterials-13-02516-f006:**
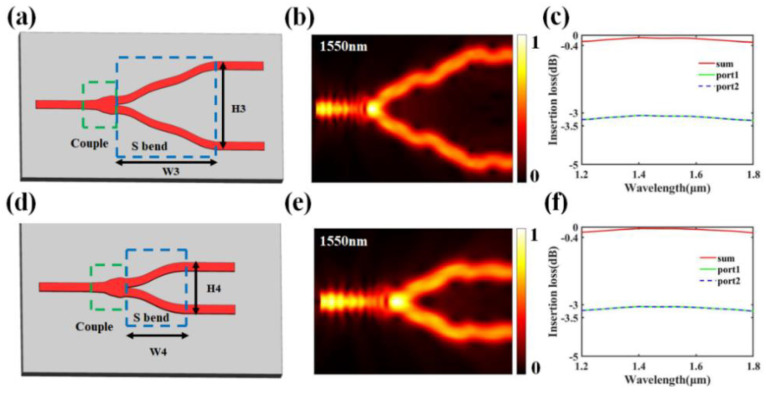
(**a**) Y-branch structure diagram of layer two in 1 × 16 OPA. (**b**) Optical field distribution diagram at 1550 nm. (**c**) Simulated insertion loss curve. (**d**) Y-branch structure diagram of layer one in 1 × 16 OPA. (**e**) Optical field distribution diagram at 1550 nm. (**f**) Simulated insertion loss curve.

**Figure 7 nanomaterials-13-02516-f007:**
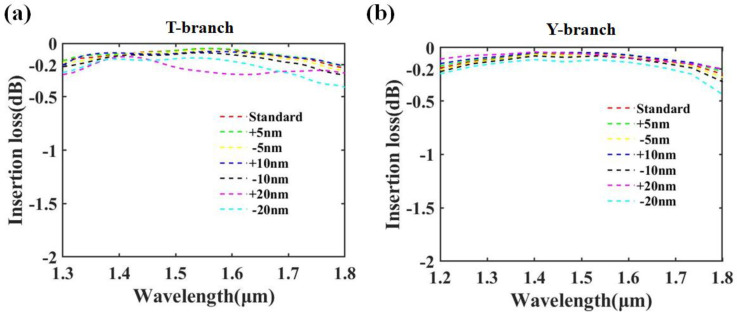
(**a**,**b**) Insertion loss with different fabrication tolerance.

**Figure 8 nanomaterials-13-02516-f008:**
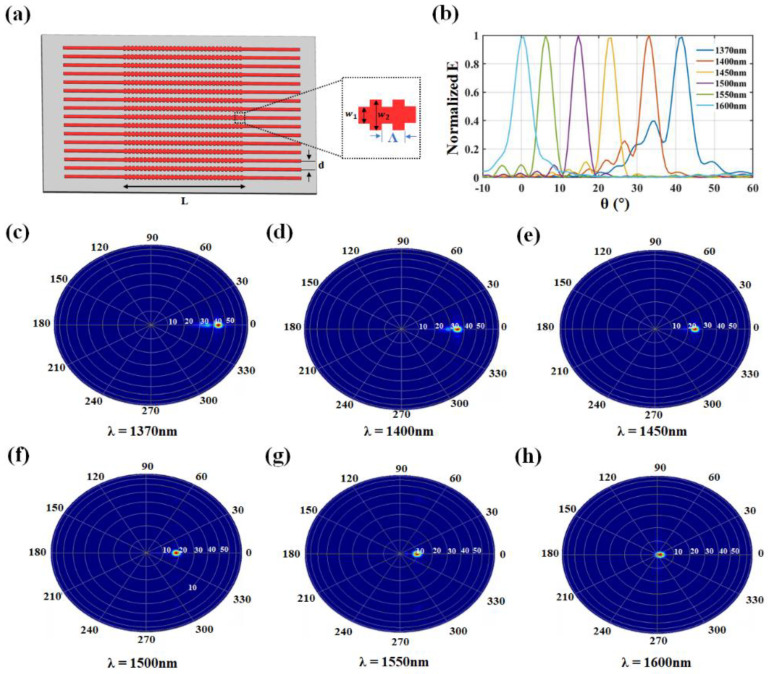
(**a**) Grating coupling structure. (**b**) Variation of diffraction angle θ in the range of 1500–1600 nm. (**c**–**h**) Far-field distributions for λ = 1370, 1400, 1450, 1500, 1550, and 1600 nm.

**Figure 9 nanomaterials-13-02516-f009:**
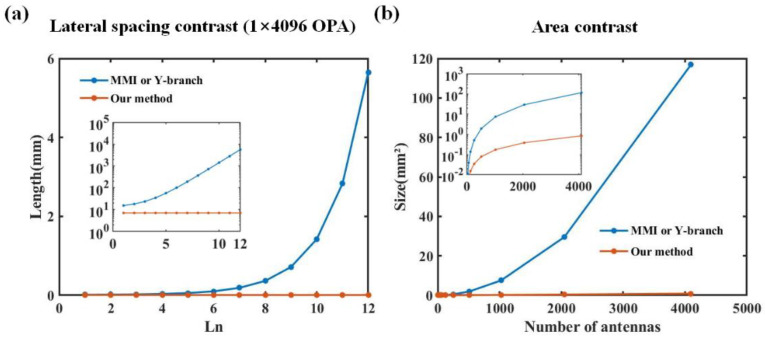
(**a**) Comparison of lateral spacing of each level in 1 × 4096 OPA. Inset: using a logarithmic scale for the *y*-axis. (**b**) Area comparison of different beam splitting schemes. Inset: using a logarithmic scale for the *y*-axis.

**Table 1 nanomaterials-13-02516-t001:** Performance parameters of T-branches and S bends.

Reference	Bend Radius (μm)	Loss (dB)	Wave Band (nm)	Type
[23]	/	0.86	1530–1570	T-branch
[43]	1.78	0.2–0.4	1550	T-branch
[44]	2.5	<0.22	1500–1600	S bend
[45]	2.5	<0.27	1400–1700	S bend
This work	~2.5	0.2	1300–1800	T-branch

Note: All comparison results are simulation results provided in the reference.

## Data Availability

Data available in a publicly accessible repository.

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
