# Peer review of "Ultra-Compact and Broadband Nano-Integration Optical Phased Array"

_nanomaterials, 2023, doi:10.3390/nano13182516_

Round 1
Reviewer 1 Report
The manuscript describes an ultra-compact broadband OPA beam-splitting scheme with nano-inverse design aimed at obtaining low-loss and reduced-dimension beam splitter structures composed entirely of inverse-designed elements.
The work is really interesting, clear, well written and, in my opinion, it is suitable for the publication. The authors demonstrated that this technique is suitable for the design of these kind of optical structures. However, in order to increase the quality of the paper, I suggest the authors addressing these few points:
1 1) The authors should better describe the section related to FDTD simulations used as a part of the design process. Are these simulations fully 3D, or do they use a 2D-equivalent approach? This is important in order to understand the complexity of the process. 3D simulations, on an optimization procedure, could be quite demanding in term of total time needed for obtaining the final results. Moreover, I think that mentioning the duration of these simulations and of the overall optimization process (specifying also in which computer architecture the simulations are performed) should be provided to the readers.
2 2) The authors should mention which FDTD code (and, if possible, also the implementation of the optimization procedure) has been used for the simulations (is it a commercial code? Do they use a home-made FDTD? How do they implement the optimization procedure?), to let the reader understanding if the optimization procedure can be implemented in general to other similar problems. I think that the contribution of this paper is important for possible applications of the described procedure to the design of other devices.
33) When presenting the results of different designs for 1x32, 1x64, 1x128, it seems that there aren’t modifications in the insertion loss (Fig. 4). This is quite expected, in my opinion, as the height of the structure allows the outputs to be decoupled from the input/divider section. Does this happen also when the T-branch is designed for the final elements of the array (1x2, 1x4), when the lateral distance is smaller and, probably, the divider and the outputs are not totally decoupled?
4 4) The authors proposed using 800nm as a width of the straight waveguide. Why this choice? The waveguide should be multimode with this width. Any complication/advantage determined by the choice of this dimension?
5) 5) In the paper, the authors mention Euler waveguides. Any reference for this waveguide?
Reviewer 2 Report
Similar research were the most popular around 20 years ago. For example in Sakai’s paper [1] was made a comparison of the state of art beam splitters, one of which (see Fig.4 [1]) is similar (but not equal) to the proposed one in the current paper. It will be very interesting to add the comparative compression of these two designs.
It look likes that the present design is slightly better due to use of the taper and the modified curve design. It is important to present some details including the curve functions and comparison of the transmitting loss white the other already used functions, including the bend with the fixed optimum radius.
It was mentioned that simulations were done by FDTD. But the most important information is omitted, including which software is used. Is it 2D or 3D FDTD.
What is the simulation grid and other parameters of the simulation? The waveguide height is also omitted thus it look likes that 2D FDTD method is used. If so, how the material and waveguide dispersion was taken into consideration?
The figure of merit (FOM) parameter is used but it is not defined.
The comparison on Fig.1 looks not very correct as for the “traditional OPA beam splitter structure” the waveguide parts with the small curve radius are put far from the beam splitter location. For the correct comparison this strongly curved waveguides must be place just in the vicinity of the splitter like is was designed in your ultra-compact OPA scheme.
It is underlined that inverse-designed beam splitters is used. I am sorry, but I never understand this term as traditionally inverse-design is used for the fiber-to-silicon wire couplers. But, I never see the inverse-design in current structure. Please, clarify the use of this term “inverse-designed beam splitters”.
This is not the first paper related to beam splitters thus it is needing to provide additional data related the practical possibility to realize this design with declared parameters by the common used technology. Thus, it hardly needs to provide the simulated data of the technology tolerance of this design in comparison to alternative designs. For example, I never believe to obtain experimentally a T-branch with a wavelength bandwidth of 500 nm (1300–1800 nm) and an insertion loss of −0.2 dB. But it will be very interesting to estimate the real parameters of this beam splitter as function of the technology tolerance.
In conclusion, this paper could be published after the major revision that may improve its quality and make it to be very interesting for the readers.
1. Sakai, A., Fukazawa, T. and Baba, T., 2002. Low loss ultra-small branches in a silicon photonic wire waveguide. IEICE transactions on electronics, 85(4), pp.1033-1038.
Round 2
Reviewer 2 Report
Thank you for the high estimate of the reviewer comments. In my understanding your paper now is strongly improved and can be publishes as it is.